# Quantifying the effect of Warmblood Fragile Foal Syndrome on foaling rates in the German riding horse population

Mirell Wobbe[1,2]*, Friedrich Reinhardt[1], Reinhard Reents[1], Jens Tetens[3,4], Kathrin F. Stock[1,2]

1 Genetic Evaluation Division, IT Solutions for Animal Production (vit), Verden, Germany, 2 Institute for Animal Breeding and Genetics, University of Veterinary Medicine Hannover (Foundation), Hanover, Germany, 3 Department of Animal Sciences, University of Goettingen, Goettingen, Germany, 4 Center for Integrated Breeding Research (CiBreed), University of Goettingen, Goettingen, Germany

* mirell.wobbe@vit.de

**Data Availability Statement:** Data cannot be shared publicly because they are of third party source. Data are available from individual German horse breeding organizations and the German Equestrian Federation (FN) for researchers who

## Abstract

Awareness of breeders of Warmblood Fragile Foal Syndrome (WFFS) increased after a widely discussed case in the USA in 2018. The hereditary connective tissue disorder, first described by a US research group in 2011 and for which a commercial genetic test exists since 2013, is caused by a point mutation in the *PLOD1* gene, inherited autosomal recessively. Extension of molecular genetic testing and reporting of test results of organized horse breeders to their studbooks implies new opportunities for analyses. In Germany, data are centrally accessible through the integrated equine data base allowing comprehensive and population-wide investigation of the role of WFFS. The objective of this study was statistical testing for associations between WFFS and reproductive performance of German riding horses and quantifying possible differences between WFFS carriers and non-carriers, also in respect of performance traits. For this purpose, covering data from 2008 to 2020 were provided by ten German studbooks, so almost 400,000 coverings and resulting foaling rates were available for multiple analyses of variance with general and mixed linear models using procedures GLM, MIXED and HPMIXED of SAS software (version 9.2). Published breeding values of stallions were used for respective comparisons of riding horse performance. Assuming a WFFS carrier frequency of 9.5–15.0% in Warmblood horses, Hardy Weinberg principle implied an expected difference of 2.4–3.7% in the foaling rates of carrier and non-carrier stallions. Our results provided statistical evidence of detrimental effects of WFFS on the reproductive performance of Warmblood horses with about 2.7% lower average foaling rate in carriers of the mutant allele than in WFFS free sires, if mated to an average mare population. Indications of favorable dressage performance of WFFS carriers were found. Reported WFFS cases indicate only the tip of the iceberg and assessing the impact of WFFS on reproduction requires consideration of premature foal losses.

meet the criteria for access to confidential data. The data underlying the results presented in the study are available from the Trakehner Verband e. V., Rendsburger Strasse 178a, 24537 Neumuenster, Germany (info@trakehner-verband. de); the Zuchtverband fuer deutsche Pferde e.V., Am Allerufer 28, 27283 Verden/Aller, Germany (info@zfdp.de); the Verband der Zuechter des Oldenburger Pferdes e.V., Grafenhorststrasse 5, 49377 Vechta, Germany (info@oldenburger-pferde.com); the Springpferdezuchtverband Oldenburg-International e.V., Grafenhorststrasse 5, 49377 Vechta, Germany (info@oldenburger-pferde.com); the Verband der Pferdezuechter Mecklenburg-Vorpommern e.V., Charles-Darwin-Ring 4, 18059 Rostock, Germany (info@pferdzuchtverband-mv.de); the Hannoveraner Verband e.V., Lindhooper Strasse 92, 27283 Verden/Aller, Germany (hannoveraner@hannoveraner.com); the Westfaelisches Pferdestammbuch e.V., Sudmuehlenstrasse 33, 48157 Muenster, Germany (info@westfalenpferde.de); the Pferdezuchtverband Sachsen-Thueringen e.V., Kaethe-Kollwitz-Platz 2, 01468 Moritzburg, Germany (info@pzvst.de); the Pferdezuchtverband Brandenburg-Anhalt e.V., Hauptgestuet 10 a, 16845 Neustadt (Dosse), Germany (neustadt@pzvba.de); the Pferdezuchtverband Baden-Wuerttemberg e.V., Am Dolderbach 11, 72532 Gomadingen-Marbach, Germany (poststelle@pzvbw.de); and the German Equestrian Federation (FN), Freiherr-von-Langen-Straße 13, 48231 Warendorf, Germany (fn@fn-dokr.de). Additionally, in order to request access to the referenced, unpublished data, S. Müller-Herbst, interested researchers may contact S. Müller-Herbst at labogen@laboklin.com. The authors confirm that others would be able to access these data in the same manner as the authors and that they did not have any special access privileges that others would not have.

**Funding:** MW has received funding from the H. Wilhelm Schaumann foundation, Hamburg, Germany. The H. Wilhelm Schaumann Foundation (http://www.schaumann-stiftung.de) is dedicated to the non-profit promotion of animal and agricultural sciences and has a special program to support young scientist under which MW has received a PhD scholarship. A grant number has not been awarded. The funders had no role in study design, data collection and analysis, decision to publish, or preparation of the manuscript.

**Competing interests:** The authors have declared that no competing interests exist.

## Introduction

Hereditary connective tissue disorders, known in many animal species, are often referred to as Ehlers Danlos Syndrome like diseases. This relates to the fact that such disorder in humans received attention of research in the late 19th century and refined descriptions were provided in the early 20th century by Edvard Ehlers and Henri-Alexandre Danlos [1–3]. Only later it was recognized that the disease belonged to a larger disease complex, then called Ehlers Danlos Syndrome (EDS), which has many different subtypes with similar clinical appearance, but different genetic background. The main symptoms are hyperextensible and abnormally fragile skin as well as hypermobility of joints. Additional findings include blood vessels prone to rupture, impaired function of internal organs and disturbed wound healing [4, 5]. In the horse, the probably best-known disease of the EDS group is the Hereditary Equine Regional Dermal Asthenia (HERDA), occurring in Quarter Horses and related breeds [6, 7]. With similar clinical appearance, but different range of affected breeds, the Warmblood Fragile Foal Syndrome (WFFS) was first described and genetically characterized in 2011 [8]. Although a genetic test for WFFS became commercially available only two years later, horse breeders have long paid little attention to the condition and opportunities of genetic testing. This has changed drastically after a case of WFFS had occurred in the US sport horse population at the beginning of 2018: An intense and often highly emotional discussion started, was carried on in the social media and quickly spread within the horse breeding sector worldwide. This has led to a substantial increase of the number of genetically tested horses, especially sport horses [S. Müller-Herbst, unpublished data].

WFFS is a connective tissue disorder that is inherited autosomal recessively, so only homozygous horses show the typical signs of the disease with, amongst others, severe lesions of the skin. Homozygous offspring can only arise from matings of two carriers, thus if a carrier sire has been mated with a carrier mare resulting in on average 50% heterozygous offspring, 25% offspring with two copies of the mutant WFFS allele and 25% offspring with two copies of the wildtype allele, so free from the mutant. Contrary to what the name suggests, the defect does not confine to Warmblood horses, but is also occurring in the Thoroughbred horse and other equine breeds [8–12]. If affected foals are born alive, they are not viable, so die soon after birth or need to be euthanized early. The genetic defect is caused by a point mutation in the equine procollagen-lysine,2-oxoglutarate 5-dioxygenase 1 (*PLOD1*, *PLOD1:c.2032G>A*, *NC_009145.3:g.39927817C>T*; OMIA [13]) gene encoding the enzyme lyslhydroxylase 1 [8]. This enzyme is responsible for the transformation of the amino acid lysine into hydroxylysine, which in turn plays an important role for the strength of collagen fibers [14, 15]. The consequences of the mutation are accordingly a thin, fragile and brittle epidermis, which is not properly connected to the subcutaneous tissue and therefore ruptures even at low stress. In addition, joints are weak, lax and hyperextensible, which is most obvious in the limbs and especially in fetlock joints [8, 16]. However, the numbers of reported cases of WFFS are worldwide relatively small, giving reason to assume that the genetic defect may also affect different earlier developmental stages, so lead to abortion, prenatal or premature foal loss, reflected in reduced foaling rates and a lack of homozygotes among the foals born. Recent results of a necropsy study using diagnostic material from abortion and stillbirth cases of horses have supported this assumption [17]. The aim of this study was to statistically test the hypothesis that foaling rates are lower in matings of carriers of the mutation than in matings of non-carriers. For these analyses we used the officially reported results of the genetic test for WFFS of stallions and broodmares together with routinely collected covering data from the German riding horse population. In addition, we used sire breeding values from the routine national genetic evaluation for riding horses in Germany to test the hypothesis that sport horses of high performance potential are over-represented among the known WFFS carriers.

## Material and methods

### Covering data

In the organized breeding of riding horses in Germany, coverings of mares are registered regardless of the mating type (natural mating or artificial insemination) and covering data of most of the studbooks are centrally managed via their joint equine data base at vit (IT Solutions for Animal Production) in Verden, Germany. For this study, covering data from that data base were made available by ten German horse breeding associations: Trakehner Verband e.V., Zuchtverband fuer deutsche Pferde e.V. (ZfdP), Verband der Zuechter des Oldenburger Pferdes e.V., Springpferdezuchtverband Oldenburg-International e.V., Verband der Pferde-zuechter Mecklenburg-Vorpommern e.V., Hannoveraner Verband e.V., Westfaelisches Pfer-destammbuch e.V., Pferdezuchtverband Sachsen-Thueringen e.V., Pferdezuchtverband Brandenburg-Anhalt e.V., and Pferdezuchtverband Baden-Wuerttemberg e.V.. Data are routinely recorded in the year of breeding until the end of the breeding season, and at least the record of the last covering of the mare in that season should always be available, because it is relevant for balancing and statistics of stallion owners and studbooks. This last record per breeding season of a given mare may be referred to as effective covering. The closed system of covering data management implies that there is no retrospective recording of coverings only after foaling which would likely cause bias towards documentation for living foals.

In the analyzed time period from 2008 to 2020, there were almost 510,000 coverings records of the participating studbooks. Only records with uniquely specified identifications of both stallion and mare were considered. Because of uncertain completeness of reporting of all subsequent coverings of individual mares into the database and to allow comparative statistical analyses across studbooks and stallion stations or semen providers, a foal case was defined as sequence of coverings of a mare in a given breeding season with the last covering potentially resulting in pregnancy. Per foal case only the last record (effective covering) was kept whereas all records referring to previous and obviously unsuccessful coverings in that same breeding season were excluded regardless of whether or not the mating partner was the same across all coverings (approximately 22% of the data).

The thinned data set underwent specific plausibility checking which included verification of each animal identification, Unique Equine Life Number (UELN), in the covering data with the official pedigree data. Gestation length was calculated from the date of birth of the foal and the last covering date, and only entries for live-born foals fitting to a realistic range of gestation lengths of 300–400 days were retained. The breeding age was calculated for both mares and stallions from covering date minus date of birth, and records indicating a covering age of less than 24 months of the stallion and/or mare were excluded as were records indicating an extremely old covering age of the mare of more than 420 months (35 years). Overall, less than 6.4% of the records were removed by that data cleaning, so the final dataset contained 395,659 records (foal case records).

The 132,188 mares relating to these coverings had on average 3.0 observations (foal cases, i. e. active breeding seasons) in the thirteen years study period. The information density per sire was considerably higher: 8,359 sires were on average represented by 47.3 foal cases, and 6,528 (78%) of them had more than one mare covered. The mean breeding ages were 10.8 years in the dams and 9.0 years in the sires, and the proportions of analyzed coverings from mares and/ or stallions younger than 3.5 years of age was smaller (6.4% in the mares, 13.7% in the stallions) than from mares and/or stallions older than 15 years of age (23.2% in the mares, 17.3% in the stallions).

The different sizes of the participating breeding associations were reflected by the variation of the amount of contributed data which ranged between 7,000 and more than 130,000

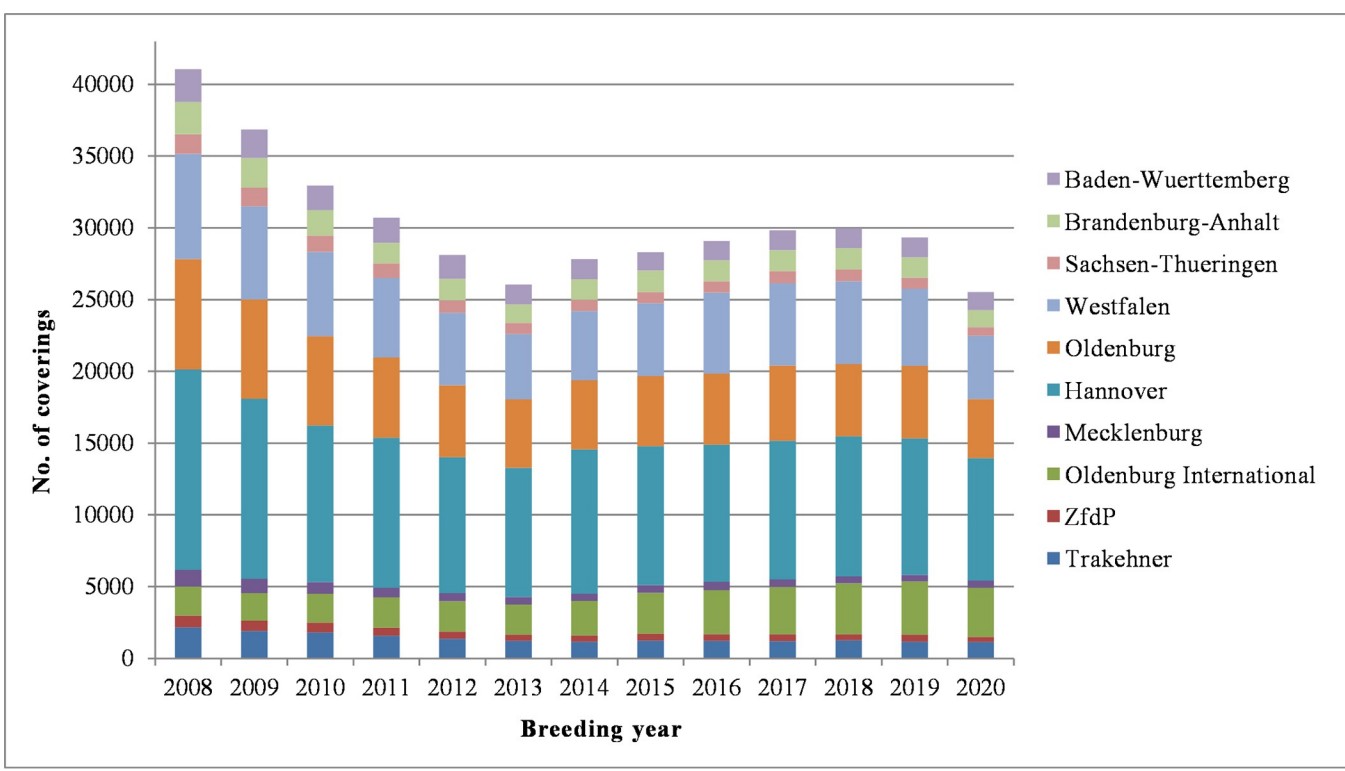

**Fig 1. Distribution of effective coverings by year and breeding association.** Covering data from ten German studbooks of riding horses in the study period from 2008 to 2020 were available, and last covering records of mares per breeding season (effective coverings) were used for analyses.

effective coverings per breeding association within the thirteen years period. The maximum annual number of effective coverings across breeding associations exceeded 40,000 and was found in the beginning of the study period (2008), whereas the minimum annual number of about 25,500 effective coverings was found at the end (2020). The distribution of the effective coverings per year and breeding association between 2008 and 2020 is shown in Fig 1.

## WFFS genotype data

Results of the commercial genetic test for WFFS (single gene test based on the research work of Winand [8]) were available for 1,982 sires (23.7% of all sires) and for 3,476 broodmares (2.6% of all mares) in the study dataset. Amount and distribution of WFFS test data reflected that testing of horses and reporting of the WFFS test results was recommended, but completely voluntary until 2019, when it became obligatory for active sires. To perform the DNA-based test, sample material (mainly hair root samples, alternatively blood or semen samples) had been sent directly to the laboratory by the breeders or stallion owners who also received the results. Studbooks invited their members to report available test results for getting a realistic picture of WFFS distribution in the population. Furthermore, several studbooks supported testing through special offers for their members, which required management of orders through the studbooks. At the same time, only very few laboratories were involved in the WFFS testing due to the legal situation (patent license for several European countries including Germany kept by Laboklin GmbH). This concept for central data collection supported assembly of a dataset with WFFS information suitable for population-wide statistical analyses. The distribution of horses with known test results by sex, year of birth and WFFS status is shown in Fig 2.

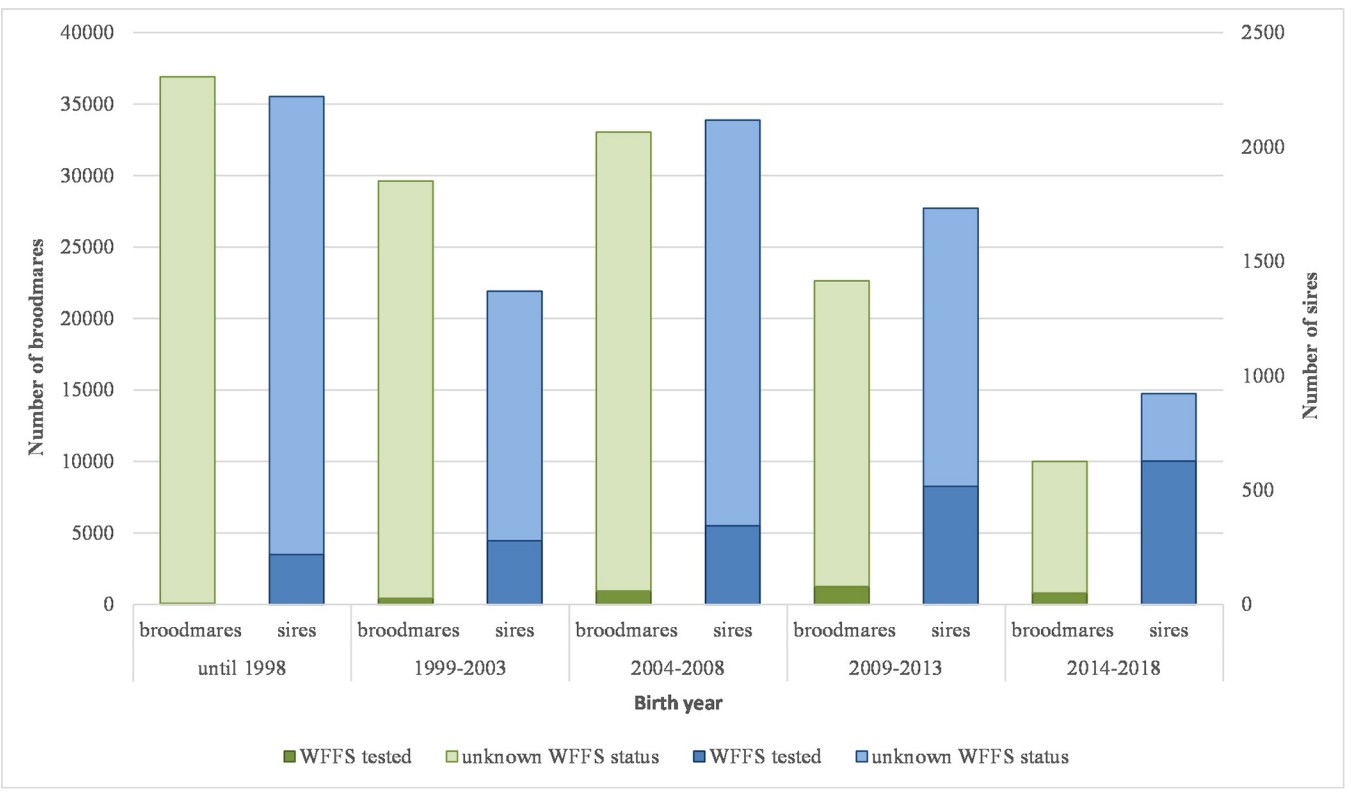

**Fig 2. Proportion of horses with known WFFS test results by sex and birth year.** The study sample included 8,359 sires (born in 1961–2018) and 132,188 broodmares (born in 1978–2018), which had records in the covering data of ten German horse breeding associations in the study period (2008–2020).

Due to the substantially lower proportion of mares with available test results, our analyses could consider only WFFS status of the sire.

## Genetic proofs of sires

The national genetic evaluation for riding horses in Germany is carried out annually on behalf of the German equestrian federation by vit and uses different data sources [18, 19]. Breeding data include performance test records of mares and stallions, national sport data contribute results of young horse competitions and regular competitions in the major riding sport disciplines, i. e. show-jumping and dressage, and international sport data contribute respective international competition results of horses registered in Germany. Pedigree data comprise all horses with records in breeding and/or sport data plus at least two ancestral generations. Individual breeding values are estimated for in total twelve traits, eight of which refer to young horse performance (JPf), two to success in national riding sport (TSP) and two to the highest level achieved (HEK), for which national and international competition results are considered. Estimations are carried out for show-jumping and dressage by Best Linear Unbiased Prediction (BLUP) using multiple trait repeatability animal models (across disciplines) for JPf and TSP and single trait animal models (within discipline) for HEK [19]. Indices are derived per discipline from the respective two (show-jumping) and six (dressage) individual JPf breeding values. All estimated breeding values (EBVs) and indices are published as relative breeding values, standardized to a mean of 100 and a genetic standard deviation of 20 points. EBVs for stallions are published when they achieve a minimum of reliability 70% and have at least five

offspring with own performance. Further details on the genetic evaluation and the publication of results for stallions are found elsewhere [18, 19].

Among the stallions with an available WFFS test result, the number of stallions with own and/or progeny records in the most recent genetic evaluation and thus breeding values from 2020 was 1,674. EBVs for JPf and TSP were available for 1,669 stallions, and EBVs for HEK were available for 1,178 stallions. According to the specific publication criteria, the numbers of stallions with published breeding values per trait group were lower and differed between disciplines: for JPf 721 in show-jumping and 850 for dressage; for TSP 433 in show-jumping and 426 in dressage; for HEK 401 in show-jumping and 382 in dressage. Most of the studbooks represented in the study have a breeding goal targeting at riding and sport horses, performing successfully in both major disciplines of riding sport, so dressage and show-jumping. The only exception is Oldenburg International, which exclusively focusses on show-jumping. The distribution of studbooks within the EBV datasets mainly reflected the size of the breeding associations, with the highest proportions of Hannover and Oldenburg, followed by Oldenburg International (slightly larger proportion in the datasets with published jumping EBVs) and Trakehner.

## Statistical analyses

Quality control of the raw data, data selection for analyses and all evaluations were performed using the SAS software package, version 9.2 (Statistical Analysis System, SAS Institute Inc., Cary, NC, USA).

**Association between WFFS and reproductive performance.** To investigate the potential effect of WFFS on the reproductive performance, a binary trait reflecting the probability of a living foal was defined: It was coded 0 if a foal case was completed without a foaling record or records indicated that the foal was born dead or died within the first two days of life; and it was coded 1 if an effective covering resulted in a successfully completed foal case, i. e. birth of a living foal and no early death of that foal was recorded (survival beyond day two of life). For the descriptive statistics, basic SAS procedures FREQ and MEANS were used. Model development and analyses of variance were performed using general and generalize linear models with the procedures GLM, MIXED and HPMIXED of SAS. Based on the model fit statistics, the following model was chosen:

$$y_{ijklmn} = \mu + Provider_i + Year_j + Age\_dam_k + WFFS\_sire_l + sire_m + e_{ijklmn}$$

with $y$ indicating the trait expression, i. e. the probability of a living foal, that was modelled using the model constant μ and the following fixed and random effects: $Provider_i$ = fixed effect of the i-th data provider (i = 1–416 unique database number of the provider of the covering data if indicated, recording breeding organization otherwise), $Year_j$ = fixed effect of the j-th year of covering (j = 1–13; individual years from 2008 to 2020), $Age\_dam_k$ = fixed effect of the k-th age of the mare (k = 1–6; age classes defined as $\leq$ 3.5 years, > 3.5 to 5 years, > 5 to 7 years, > 7 to 10 years, > 10 to 15 years, > 15 years), $WFFS\_sire_l$ = fixed effect of the l-th WFFS status of the sire (l = 1–3; free, carrier, unknown), $sire_m$ = random effect of the m-th sire (m = 1–4,836; individual sires if represented by at least 5 foal cases, dummy sire for the others), and $e_{ijklmn}$ = random residual.

For the analyses of variance, we used either the full data or subsets of the data resulting from restrictions regarding WFFS status information on the sire and minimum information per sire and/or provider. The distribution of studbooks in these datasets (S1 Table) was similar to the distribution in the whole covering data shown in Fig 1. Details on the six datasets used for the statistical analyses can be seen from Table 1.

**Table 1. Characteristics of the datasets used for the statistical analyses of covering data regarding WFFS effects from 10 German studbooks of riding horses, in dependence of the applied restrictions.**

| Dataset | Restrictions | | Characteristics | | | | |
|---------|--------------|--|-----------------|--|--|--|--|
| | Inclusion criteria for sires and/or CDP | WFFS status information | No. of sires | Distribution of sires by WFFS status (in %) | | No. of CPD | Mean No. of sires (coverings) per CPD | No. of effective coverings (mares) |
| 1a | No | No | 8,359 | C 149 | (1.8) | 416 | 67 (882) | 395,659 |
| | | | | F 1,833 | (21.9) | | | (132,188) |
| | | | | U 6,377 | (76.3) | | | |
| 1b | No | Yes (sire: C, F) | 1,982 | C 149 | (7.5) | 363 | 76 (997) | 262,513 |
| | | | | F 1,833 | (92.5) | | | (98,932) |
| 2a | Yes (sires with ≥ 10 foal cases) | No | 3,655 | C 113 | (3.1) | 409 | 68 (894) | 381,287 |
| | | | | F 1,292 | (35.3) | | | (127,336) |
| | | | | U 2,250 | (61.6) | | | |
| 2b | Yes (sires with ≥ 10 foal cases) | Yes (sire: C, F) | 1,405 | C 113 | (8.0) | 360 | 77 (1,002) | 260,267 |
| | | | | F 1,292 | (92.0) | | | (98,094) |
| 3a | Yes (CDP with direct electronic reporting) | No | 2,899 | C 94 | (3.2) | 406 | 21 (475) | 221,470 |
| | | | | F 1,100 | (37.9) | | | (82,097) |
| | | | | U 1,705 | (58.8) | | | |
| 3b | Yes (CDP with direct electronic reporting) | Yes (sire: C, F) | 1,194 | C 94 | (7.9) | 353 | 24 (532) | 163,833 |
| | | | | F 1,100 | (92.1) | | | (66,961) |

CDP = provider of covering data, F = free of WFFS (no WFFS defect allele), C = carrier of the WFFS defect allele, U = WFFS status unknown.

To facilitate interpretation of the results of the analyses of variance, especially the differences of least square mean (LSM) estimates for the WFFS status of the sire, the expected proportion of homozygous progeny from a carrier sire was calculated using the Hardy-Weinberg equilibrium (HWE):

$$p^2 + 2pq + q^2 = 1$$

with allele frequencies p and q, and assuming random mating and frequencies of the WFFS defect allele of 9–15% in the recent riding horse population [8, 20; S. Müller-Herbst, unpublished data].

**Association between WFFS and sport performance.** To evaluate WFFS genotype effects on the sires' performance potential for riding sport, we used sire EBVs from the 2020 routine national genetic evaluation for riding horses in Germany. As for the covering data, basic SAS procedures FREQ and MEANS were used for the descriptive statistics. Model development and analyses of variance were performed using general linear models with the procedure GLM of SAS. The final model was:

$$y_{ijkl} = \mu + pEBV_i + WFFS\_sire_k + e_{ijkl}$$

with $y$ indicating the trait expression, i. e. the EBV of the sire with available foal case data, that

was modelled using the model constant μ and the following fixed and random effects: $pEBV_i$ = fixed effect of the i-th paternal EBV (i = 1–1,442), $WFFS\_sire_k$ = fixed effect of the k-th WFFS status of the sire (k = 1–2; free, carrier), and $e_{ijkl}$ = random residual.

For the analysis of variance, we considered stallions with known WFFS status and used two datasets: one including all stallions with breeding values, and the other one including only those stallions with published ones.

## Results

### Covering data

The analyses of variance revealed significant influences of the age of the broodmare and of the WFFS status of the sire on the probability of a living foal in the considered fraction of the German Warmblood population. Broodmares at the age of 10 to 15 years and older had a markedly decreased chance of a successful birth (in average around 0.50 in comparison to 0.65–0.70 for younger mares). For known WFFS carriers, the LSM estimates for the probability of a living foal were consistently lower than for the sires tested WFFS free. The LSM differences ranged between 0.025 and 0.030 and were significant with P < 0.01 when considering only sires with known WFFS status in the analyses (datasets 1b, 2b, 3b). Without the restriction regarding the WFFS status of the sire (datasets 1a, 2a, 3a), the LSM differences between WFFS free and carrier sires were slightly lower (0.021–0.029) but remained significant for two of the three analyzed datasets (dataset 2a and 3a) (Table 2).

The WFFS carrier frequencies per studbook found in our data were similar to the reported frequencies [8, 20, S. Mueller-Herbst, unpublished data] and ranged from 2.5–13.5% with highest figures for Hannover and Oldenburg and lowest figures for Trakehner. Based on an assumed frequency of carriers (heterozygotes) of the WFFS mutation of between 9 and 15%, the expected proportion of homozygous progeny from a carrier sire calculated using HWE was 2.4–3.7%. In our data (dataset 1a), we found a carrier frequency of 9.15% among all horses with known WFFS status. This figure from the German riding horse population was accordingly at the lower end of the frequency range found in the literature for the Warmblood and used for the comparative HWE calculation of expected differences.

### Genetic proofs of sires

WFFS status was neither in all stallions with EBV nor in those stallions with published EBV significant for any of the six EBVs relating to show-jumping (P > 0.05). In those cases, where P values approached significance, trends of higher or lower EBVs in carriers than in non-carriers were non consistent across the two groups of stallions analyzed. Conversely, WFFS status

**Table 2. Results of the multiple analyses of variance for the probability of a living foal and the WFFS status of the sire, with least square mean estimates (LSM) and differences with respective standard errors (SE) and error probabilities (P).**

| Dataset | LSM (SE) for WFFS status | | | P | LSM difference C vs. F (SE) | P Diff$_{C-F}$ |
|---|---|---|---|---|---|---|
| | Free (F) | Carrier (C) | Unknown | | | |
| 1a | 0.624 (0.006) | 0.603 (0.012) | 0.624 (0.006) | 0.16 | -0.021 (0.011) | 0.06 |
| 1b | 0.631 (0.006) | 0.606 (0.010) | - | <0.01 | -0.025 (0.009) | <0.01 |
| 2a | 0.622 (0.006) | 0.598 (0.012) | 0.612 (0.006) | 0.02 | -0.024 (0.011) | 0.03 |
| 2b | 0.628 (0.006) | 0.601 (0.010) | - | <0.01 | -0.028 (0.009) | <0.01 |
| 3a | 0.633 (0.006) | 0.604 (0.011) | 0.602 (0.006) | <0.001 | -0.029 (0.010) | <0.01 |
| 3b | 0.632 (0.007) | 0.602 (0.010) | - | <0.001 | -0.030 (0.008) | <0.001 |

For definition of the datasets and details on their structure see Table 1.

**Table 3. Results of the multiple analyses of variance for the estimated breeding values (EBVs) for dressage and show-jumping and the WFFS status, with number of horses in total (N) and by WFFS status ($N_F$, $N_C$), least square mean estimates (LSM) with respective standard errors (SE) and error probabilities (P) for all stallions with EBVs (all) and for the stallions with published EBVs (pub).**

| Discipline and EBV | | N horses ($N_F$, $N_C$) | LSM EBV (SE) | | P |
|---|---|---|---|---|---|
| | Group of stallions | | Free (F) | Carrier (C) | |
| Show-jumping | | | | | |
| free jumping | all | 1,669 (1,542, 127) | 98.00 (0.24) | 99.42 (0.84) | 0.11 |
| | pub | 721 (650, 71) | 104.03 (0.42) | 105.90 (1.27) | 0.16 |
| jumping under rider | all | 1,669 (1,542, 127) | 100.39 (0.22) | 101.76 (0.76) | 0.08 |
| | pub | 721 (650, 71) | 104.52 (0.38) | 105.41 (1.15) | 0.46 |
| ABP show-jumping | all | 1,669 (1,542, 127) | 101.94 (0.23) | 102.83 (0.81) | 0.29 |
| | pub | 721 (650, 71) | 104.81 (0.40) | 104.29 (1.20) | 0.68 |
| JPf index show-jumping | all | 1,669 (1,542, 127) | 100.50 (0.26) | 101.94 (0.89) | 0.12 |
| | pub | 721 (650, 71) | 105.32 (0.44) | 105.97 (1.34) | 0.64 |
| TSP show-jumping | all | 1,669 (1,542, 127) | 101.97 (0.23) | 101.23 (0.82) | 0.38 |
| | pub | 433 (382, 51) | 110.48 (0.58) | 107.38 (1.58) | 0.07 |
| HEK show-jumping | all | 1,178 (1,074, 104) | 100.98 (0.40) | 99.63 (1.27) | 0.31 |
| | pub | 401 (350, 51) | 111.41 (0.81) | 107.10 (2.13) | 0.06 |
| Dressage | | | | | |
| walk | all | 1,669 (1,542, 127) | 104.77 (0.25) | 107.37 (0.86) | <0.01 |
| | pub | 850 (768, 82) | 107.35 (0.37) | 108.77 (1.12) | 0.23 |
| trot | all | 1,669 (1,542, 127) | 105.37 (0.27) | 109.45 (0.95) | <0.001 |
| | pub | 850 (768, 82) | 108.65 (0.41) | 111.30 (1.26) | 0.05 |
| canter | all | 1,669 (1,542, 127) | 105.52 (0.26) | 109.32 (0.92) | <0.001 |
| | pub | 850 (768, 82) | 107.13 (0.39) | 109.10 (1.19) | 0.11 |
| rideability | all | 1,669 (1,542, 127) | 105.06 (0.26) | 108.57 (0.90) | <0.001 |
| | pub | 850 (768, 82) | 107.40 (0.38) | 109.37 (1.16) | 0.10 |
| ABP dressage | all | 1,669 (1,542, 127) | 106.13 (0.29) | 109.24 (0.93) | <0.01 |
| | pub | 850 (768, 82) | 108.27 (0.40) | 109.54 (1.22) | 0.32 |
| JPf index dressage | all | 1,669 (1,542, 127) | 106.67 (0.30) | 110.64 (1.04) | <0.001 |
| | pub | 850 (768, 82) | 109.48 (0.44) | 111.45 (1.35) | 0.16 |
| TSP dressage | all | 1,669 (1,542, 127) | 104.92 (0.23) | 105.79 (0.81) | 0.30 |
| | pub | 426 (375, 51) | 108.89 (0.62) | 108.15 (1.68) | 0.68 |
| HEK dressage | all | 1,178 (1,074, 104) | 102.19 (0.39) | 103.38 (1.25) | 0.37 |
| | pub | 382 (333, 49) | 112.10 (0.91) | 111.59 (2.36) | 0.84 |

was significant for six of the eight dressage-related EBVs, including all EBVs for young horses performance in the discipline dressage, when considering all stallions (P < 0.01). For these traits and in that stallion sample, LSM differences ranged between 2.60 and 4.08 and implied higher genetic potential of WFFS carriers with regard for dressage performance. When only stallions with published EBVs were considered, the pattern of results remained the same as in all stallions with EBVs, while P values increased. Tendencies of genetic superiority of WFFS carriers in dressage performance of young horses were still identified for trot and rideability (P ≤ 0.10; Table 3).

## Discussion

This study was based on the routinely recorded covering data of ten German horse breeding associations over a thirteen years period. According to the national annual summary statistics as published by the German national federation, the contributing associations ensured

coverage of a large part of Warmblood breeding in Germany [21]. Furthermore, the distribution of the numbers of coverings per year and association in our data reflected the overall development of horse breeding in Germany (Fig 1). Previous studies about the population structure in German Warmblood breeds have shown high genetic similarities and many cross-links between the different breeds exists [22]. Accordingly, there were no indications of systematic bias relating to the data selection for this study, and the results may be considered representative for the whole German riding horse population. Possible breed differences were considered in the model via the fixed effect of the data provider, having a unique database number for each studbook.

Because recorded covering data are heterogeneous in terms of their completeness, we performed our analyses based on effective coverings and with focus on the probability of a living foal. Complete coverage sequences would allow investigating alternative fertility trait definitions like mating success or number of coverings (inseminations) per breeding season. However, this would require an equally high level of data completeness across studbooks and data sources and consistency of the covering reports over time. Subsets of data for which documentation of all coverings could be verified may be used for future in-depth study of the effects of WFFS on equine fertility and reproductive success.

The reproductive performance of a broodmare is influenced by a variety of factors, many of which are difficult to capture and usually remain unavailable for systematic analyses [23]. The overall level of reproductive success is around 50–60% for Warmblood mares, with best results for pasture breeding and artificial insemination (AI) with fresh semen, while the most common type for breeding in Warmblood horses is AI with cooled shipped semen [23]. Besides the mating type (mating in hand, AI with fresh, cooled or frozen semen, embryo transfer, etc.), which is of course also related to the stallion and not only to the mare, the mare's reproductive history and the specific environmental conditions in the breeding period, relating to time of the year, light exposure, climate and also training status or alternative use of the broodmare have been described as relevant for the individual mare's breeding success [23]. However, the lack of additional information in the routinely collected covering data of the studbooks limited optimization options during model development and regarding the choice of fixed and random effects for the final model.

Age is known to play an important role for the reproductive performance of a broodmare, and reduced fertility of older mares (over 15 years old) is described in the literature [23, 24] and was also reflected in our data. A strong influence of the broodmare's age on her reproductive success was found, so age of the mare was included in the model as a fixed effect with 6 levels. After finishing the model design, we created then sub datasets with different restrictions which may influence the outcome (minimum number of foals per sire on the one hand and only records which were directly electronical reported on the other hand) in order to check if the results stay consistent.

Analyses of variance showed significant differences between matings with carriers of the WFFS mutant allele and WFFS-free sires with regard to the probability of a living foal. Higher differences could be found in the sub datasets excluding the sires with unknown WFFS status (1b, 2b, 3b). We did not exclude these sires in total from the analyses to investigate also the LSM results for them, expecting values between those for LSM of WFFS carrier and free sires, which was the fact in the datasets 2a and 3a with significant p-values, thus reflecting a group of stallions in which both statuses occur. Significant and marked differences were found in the dataset including only data providers with a direct electronic reporting to the database system. Analyses of the proportions of living foals in the datasets including only providers with direct electronic reporting revealed largest LSM differences with smallest SE, so clearest patterns of reproductive performance of mares in dependence on the WFFS status of the sire. This

improved discrimination between the sires by their WFFS status in the presumably most complete data sets (3a, 3b) is in agreement with a relationship between the way of reporting and data completeness and accordingly supports the assumption that successful coverings were more likely to be entered to the data base through alternative ways than those which did not result in the birth of a foal. Although it may be very difficult to counteract the effect of the reproductive success on the availability of data, our results indicate clear benefits for usage of data, that is electronically reported and as complete as possible, for monitoring purposes and possibly further statistical analyses.

The statistically significant differences between the sire categories ranged from 0.024 to 0.030, so were very similar to the expected proportion of homozygous offspring from a mating of carrier sire to the average mare population (0.024–0.037). The latter was calculated using HWE and literature values for the frequency of the WFFS defect allele, represented in the Warmblood horse population [8, 20; S. Müller-Herbst, unpublished data]. Nevertheless, the frequency of the WFFS defect allele found in our data was at the lower end of the range of figures found in the literature which may also due to the fact, that the reporting of the WFFS results is voluntary for mares and the breeders may be more likely to report test results of WFFS-free broodmares to their studbook. Because of this much more mares in our data were not tested for WFFS than it was the case for the stallions, so our calculations were referring to the sires. With more complete WFFS data also for the mares further studies can address the foaling rates from matings where WFFS status for both parents is known. The assumption that WFFS is also a reason for prenatal foal losses and so causes a reduced chance of a living foal by 0.025 to 0.030 if mating with a carrier sire, could be confirmed in this study. The pathophysiology of WFFS and previously reported observations [16, 17] suggested that prenatal foal losses may also occur due to this inherited defect. The results of our study supports this reasoning and provided statistical evidence that prenatal foal losses relate to WFFS (and not only abortion or birth of a not viable foal). Therefore, horse breeders should be aware of the WFFS status of at least one and preferably both of the suggested mating partners. A term of further studies with focus on the histological findings concerning WFFS could be a conceivable correlation in the prenatal foal losses with the enzyme function, which is encoded from the affected *PLOD1* gene and which is essential for the strength of collagen fibers [14, 15] and thus may play a role for discontinuity of connection between mare and foetus while pregnancy. Other studies indicate that the WFFS mutation may more likely affect later than earlier stages of foal development [17, 25]. Embryonic survival may therefore be an interesting field for further study to validate previous findings.

Based on the study of the WFFS mutation in a wider range of horse breeds and populations, it has been suspected that the first occurrence of the WFFS mutation dates back several centuries and occurred for the first time in an early ancestor with substantial influence on multiple lines of breeds [9, 25]. Although reproductive success may not belong to the major breeding goals of most breeds of horses, the question arises why the detrimental effect of the WFFS mutation on reproductive performance has not taken effect according to basic evolutionary principles and promoted the loss of the mutant allele. From an evolutionary point of view, the relatively high frequency of around 10% WFFS carriers in different populations of riding horses, determined nowadays, so possibly centuries after the first occurrence of the mutation, implies that the reproductive disadvantage acting towards frequency reduction must have been counterbalanced by selection pressure acting in the opposite direction. Favorable aspects linked with the mutation may accordingly explain the long survival of the defect allele. Such involuntary co-selection resulting in a correlated selection response and respective stable or even increasing carrier frequencies in organized breeding, has been described, for example, for Polysaccharide Storage Myopathie 1 (PSSM1) in the Quarter Horse and draft horse breeds

[26–28]. The genetic defect PSSM1 causes alterations in carbohydrate metabolism linked to certain types of muscle fibers which lead to degenerative muscle diseases in homozygotes but pronounced muscles and superiority in long distance endurance in heterozygotes. In the mentioned breeds of horses, the heterozygous individuals had a clear selection advantage, such that the PSSM1 mutation was retained, and its frequency stabilized at higher level. It is also found in several other horse breeds worldwide, but apparently in clearly lower frequencies.

Concerning WFFS in the Warmblood horse, it has been speculated that it is especially the lines of very successful sport horses in which higher WFFS carrier rates are found. To verify such relationship between WFFS and performance, we used the WFFS test results from this study and breeding values of sires from the routine genetic evaluation for German riding horses. We were able to identify a significant relationship with the dressage performance of young horses, i.e. quality of gaits and rideability, with on average higher EBVs in WFFS carriers than in WFFS-free horses. The distributions of studbooks in the EBV dataset 'all' and 'pub' were very similar, with the only difference that in the 'pub' datasets for show-jumping there was a slightly higher proportion of Oldenburg International, consistent with its breeding focus on show-jumping horses. We suggest that the reduced number of horses in the 'pub' datasets (factor of 2.0 to 3.9) is the main reason for not achieving statistical significance (impaired statistical power). However, the patterns of results were consistent regardless of whether or not accessibility of published (official) EBVs was required. These findings are in agreement with a previous study in which breeding values for conformation and performance from one of the studbook-specific genetic evaluations for riding horses in Germany were used [25] and with a study with breeding values for Swedish Warmblood Horses [29]. The genetic superiority of carriers of the WFFS mutation with regard to dressage traits of young riding horses and the important role of dressage breeding in the majority of the breeding organizations for riding horses may accordingly be the reason for the surmised correlated selection response on WFFS. Currently data for distinguishing between possible reasons like drift, causal effects or hitchhiking with a linked locus for the impact of WFFS on the dressage performance are lacking, so that further studies are needed.

Another aspect that may have prevented elucidation and direct selection against WFFS in horse breeding for a long time may have been the partially hidden acting of WFFS through premature foal losses, which has not been noticed among other abortion causes (which are better known and play a more important role in horses, e.g. equine herpes virus type 1 and 4, equine viral arteritis). There are also cases of aborted foals, homozygous for WFFS, which do not show any of the typical symptoms and can therefore only be assigned to this genetic defect with a genetic test or histological examinations [17]. Since WFFS has been recognized as occurring more often than just sporadically in Warmblood horses and also horses of other breeds, scientists and breeding organizations have underlined their recommendations to use available test offers also for mares in order to take WFFS into account when planning mating. Active breeding stallions are required to be tested for the WFFS mutation in the German population of riding horses since 2019, and similar regulations exist in other countries. However, only with known WFFS status of both mating partners responsible decisions can be taken which also consider longer-term effects of population-wide breeding measures against WFFS. As long as mating of two carriers are consequently avoided, no homozygous offspring can develop, so the breeding is safe and precludes losses of pregnancies and foals due to WFFS. At the same time, there is no restriction of the gene pool as it would be the case if carriers of the mutant alleles would be prohibited to continue breeding. This latter approach was temporarily discussed to achieve faster reduction of the frequency of the WFFS mutation in the worldwide riding horse population.

According to the results of this study confining breeding to WFFS-free broodmares and stallions may increase the foaling rates by around 2.7%, corresponding to an improvement of reproductive success by about 4%. Genetic testing allows management of genetic defects through a responsible and targeted breeding use of carriers. The positive characteristics of these horses can then be preserved for the breeding program and development of homozygous offspring can be avoided, as it is already practiced, for example, with PSSM1.

## Conclusions

Statistical analyses of routinely collected covering data of German riding horses showed significant differences between sires in dependence on their WFFS status, with the probability of a living foal being reduced by about 2.7% in mating with WFFS carrier sires when compared to sires without the mutant allele. This supports the assumption that WFFS is a relevant cause of premature foal losses in the riding horse population and the role of WFFS is underestimated if only reported cases (affected foals) are referred to. The relatively high frequency of the defect allele implies crucial importance of genetic testing and responsible use of breeding animals. The autosomal recessive inheritance of WFFS facilitates management of the disease by strictly avoiding mating of two carriers, while keeping valuable broodmares and sires regardless of their carrier status in the overall gene pool.

In the future, increase of large-scale genotyping and genomic applications will facilitate identifying and managing causal variants also in horses. However, increase of knowledge in the field of genetic characteristics is crucially dependent on the availability of respective phenotypic data, so lessons learned from the WFFS scenario in the Warmblood should be that it is essential to have a transparent and strong data base system with systematic recording of pedigree, phenotypic and genotypic data. Relevant information (e.g., breeding records, foaling data, health events, results of genetic tests) should ideally be made by all stakeholders and provide the basis of integrated analyses. Data security and data privacy issues must be carefully considered to achieve the level of transparency which allows appropriate handling of known defects in terms of responsible breeding.

## Supporting information

**S1 Table. Studbook distribution in % within the datasets used for the statistical analyses of covering data regarding WFFS effects.**
(XLSX)

## Acknowledgments

The German studbooks and the national German Equestrian Federation (FN) which have contributed data are acknowledged for their support of this work.

## Author Contributions

**Conceptualization:** Mirell Wobbe, Friedrich Reinhardt, Jens Tetens, Kathrin F. Stock.

**Data curation:** Mirell Wobbe, Kathrin F. Stock.

**Formal analysis:** Mirell Wobbe.

**Funding acquisition:** Kathrin F. Stock.

**Investigation:** Mirell Wobbe.

**Methodology:** Mirell Wobbe, Friedrich Reinhardt, Kathrin F. Stock.

**Project administration:** Kathrin F. Stock.

**Resources:** Reinhard Reents, Kathrin F. Stock.

**Supervision:** Kathrin F. Stock.

**Validation:** Kathrin F. Stock.

**Visualization:** Mirell Wobbe.

**Writing – original draft:** Mirell Wobbe.

**Writing – review & editing:** Friedrich Reinhardt, Reinhard Reents, Jens Tetens, Kathrin F. Stock.

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
