## [Decision Letter · Decision Letter 0]

5 Jan 2022

PONE-D-21-32388Quantifying the effect of Warmblood Fragile Foal Syndrome on foaling rates in the German riding horse populationPLOS ONE

Dear Dr. Wobbe,

Thank you for submitting your manuscript to PLOS ONE. First of all, I would like to apologize for the unusually long time it took to come back to you with reviews of your manuscript, mostly due to an extended process of finding an editor. Thank you for being patient.After careful consideration, we feel that it has merit but does not fully meet PLOS ONE’s publication criteria as it currently stands. Therefore, we invite you to submit a revised version of the manuscript that addresses the points raised during the review process.

We look forward to receiving your revised manuscript.

Kind regards,

Sylvie Mazoyer, Ph.D.

Academic Editor

PLOS ONE

Journal Requirements:

Reviewers' comments:

Reviewer's Responses to Questions

**Comments to the Author**

1. Is the manuscript technically sound, and do the data support the conclusions?

Reviewer #1: Yes

Reviewer #2: Partly

2. Has the statistical analysis been performed appropriately and rigorously? 

Reviewer #1: Yes

Reviewer #2: N/A

3. Have the authors made all data underlying the findings in their manuscript fully available?

Reviewer #1: Yes

Reviewer #2: No

4. Is the manuscript presented in an intelligible fashion and written in standard English?

Reviewer #1: Yes

Reviewer #2: Yes

5. Review Comments to the Author

Reviewer #1: The manuscript by Wobbe et al. reports on an investigation of Warmblood Fragile Foal syndrome-genotype related foaling rates in German riding horses. Data from genetic testing of stallions and mares used for breeding were put into context with effective coverings per breeding season.

The manuscript is interesting and original. Some explanations and conclusions need further explanations: It is highly important to dissect the different terms used with regard to “foaling rates”: premature, prenatal and abortion do not define the same developmental stage. Furthermore, previous authors suggestions about WFFS-heterozygous genotype-associations with latter stages of gestation (Aurich, Christine, et al. "Characterization of abortion, stillbirth and non-viable foals homozygous for the Warmblood Fragile Foal Syndrome." Animal reproduction science 211 (2019): 106202.) or embryonic development (Metzger, Julia, et al. "Hanoverian F/W‐line contributes to segregation of Warmblood fragile foal syndrome type 1 variant PLOD1: c. 2032G> A in Warmblood horses." Equine veterinary journal 53.1 (2021): 51-59.) should be taken into account. The limitations of this study for estimating foal loss based on effective coverings data should be clearly stated.

In addition to that, I think that the abstract section needs to be improved. It is most essential to make clear, that a genetic carrier does not have an increased probability of lower foaling rate in comparison to the wild type genotype as long as no mating with a genetic carrier is performed. This should be specified in the abstract. Did the authors take pedigree data into account and test for foaling rates of carrierxcarrier carrierxwild type? This is an important aspect and need clarification.

Minor comments:

Line 60-61: Citation missing. It there any evidence confirming the increase of genetic testing?

Line 365-368: Citation incorrect? In this patent description comments about the earlier ancestor are missing!? Please give details where to find it or correct citation.

Reviewer #2: In the manuscript entitled ‘Quantifying the effect of Warmblood Fragile Foal Syndrome on foaling rates in the German riding horse population’ the authors investigate the association of the autosomal recessive nonsense mutation in the PLOD1 gene, causative for an Ehlers Danlos like syndrome in Warmblood horses (WFFS), with reproductive performance and estimated breeding values in German warmblood horses. Therefore, they used published data on coverings and foaling rates, estimated breeding values for twelve performance traits and published WFFS genotypes. The study involves a comprehensive dataset comprising information from 10 German Warmblood breed registries and a timespan of thirteen years.

The authors conclude, that their findings provide evidence that the segregating WFFS allele contributes to lower foaling rates (breeding success) in the German warmblood population. Second, they propose an association of the WFFS genotype (heterozygous-versus free) with dressage performance traits. There were issues with missing/incomplete data, which the authors tried to overcome.

The aim of the research was formulated precisely. However, some assumptions and calculations were not fully transparent to me. I point these out in my review below and ask for clarification, before I would give a recommendation on whether this article is suitable for publication.

Major Points:

Impact of population substructure

For their analysis, the authors treat the German Riding horse as a single population. Does this definitely apply for the dataset used? As pointed out in the methods, data from 10 different registries are used. Number of data entries differed significantly between registries (Figure 1). For my knowledge breeding goals, and breeding stock (although studbook is more or less open) differ between German Warmblood breeds. My question is: Is there population substructure in the dataset? And to what extent does the population substructure/breed registry affect the WFFS allele frequency and hence the outcome of the analysis? The authors mention that pedigree information is available for all horses used in the study.

Also, for the estimates of association of the WFFS allele and breeding value: the sampling could affect the outcome and should be considered. Association with gait EBV is just detectable in the ‘all’ dataset (Table 2) and it is not in the ‘pub’. What is the difference between ‘all’ and ‘pub’ – with regard to breeds, breeding goals/populations ?

Hardy Weinberg estimates

(L33; L263). Under the assumed carrier frequency of 9-15%, assumed by the authors, the allele frequency of the minor allele should be 0.045-0.0725. Under Hardy Weinberg equilibrium 0.002-0.005 (0.2-0.5%) homozygous offspring are expected.

Furthermore, it has recently been shown that WFFS allele frequency differs between Warmblood breeds/registries (doi: 10.3390/genes11121518), and seems to be highest in Hanoverians (doi: 10.1111/evj.13271). See above comment.

Scenarios underlying the significant impact of WFFS allele on foaling rate

The effect of the WFFS allele on foaling rates is rather minor, although significant. I would appreciate a comprehensive discussion on the scenarios that could lead to this observation.

The significant impact of WFFS allele on dressage performance

Also for this fact, a broader discussion on the possible causes (WFFS causative, hitchhiking with a linked locus or even just drift?) that could lead to the association of the WFFS allele with dressage performance would be beneficial.

Minor Points:

The title does not fully reflect the content of the article

Abstract

L27 ..also in respect ‘to other traits’. Not precise.

L33. … Hardy Weinberg principle implied an expected difference of 2.4-3.7% in the foaling rates of carrier and non-carrier stallions … This calculation is for me not clear (see major comments)

Introduction

The WFFS mutation which is central in this article should be described in more detail (Genome coordinates, OMIA link) Further the mode of inheritance, which is autosomal recessive - should be pointed out in the introduction

Methods

L161-163 fits better to section covering data

L229 allele frequency was also estimated in this work – and it was lower, why not refer to this as well?

L243 more explanation is necessary for the describing all/pub data (maybe supplementary Table)

Results

Table1: A supplementary table with detailed information on registries/breeds enclosed in the respective restriction sets would be helpful

Table2: More detailed description on breeding goals (discipline) and registries for the horses would be helpful. Were all EBVs determined in the same individuals? How can one explain loss of significance in Dressage values when only pub data are analysed?

Discussion

The discussion could be shortened, more critically reflecting the results of this work.

L334 This conclusion is not clear to me.

6. PLOS authors have the option to publish the peer review history of their article (what does this mean?). If published, this will include your full peer review and any attached files.

Reviewer #1: No

Reviewer #2: No

---

## [Author Response · Author response to Decision Letter 0]

3 Apr 2022

Reviewer 1:

It is highly important to dissect the different terms used with regard to “foaling rates”: premature, prenatal and abortion do not define the same developmental stage.

To avoid misinterpretation, the respective section in the introduction where reference is made to lower foaling rate has been adjusted (ll. 80-82).

Furthermore, previous authors suggestions about WFFS-heterozygous genotype-associations with latter stages of gestation (Aurich, Christine, et al. "Characterization of abortion, stillbirth and non-viable foals homozygous for the Warmblood Fragile Foal Syndrome." Animal reproduction science 211 (2019): 106202.) or embryonic development (Metzger, Julia, et al. "Hanoverian F/W-line contributes to segregation of Warmblood fragile foal syndrome type 1 variant PLOD1: c. 2032G> A in Warmblood horses." Equine veterinary

journal 53.1 (2021): 51-59.) should be taken into account.

The suggested studies indicating that the WFFS mutation may more likely affect later rather than earlier stages of foal development are now referred to in the discussion of the potential pathological effects (ll. 389-392).

The limitations of this study for estimating foal loss based on effective coverings data should be clearly stated.

In the discussion, we clearly address the study limitations and also indicate options for subsequent work helping to validate the results (ll. 318-325). 

In addition to that, I think that the abstract section needs to be improved. It is most essential to make clear, that a genetic carrier does not have an increased probability of lower foaling rate in comparison to the wild type genotype as long as no mating with a genetic carrier is performed.

The abstract has been revised such that it is becoming clearer that also with regard to lower foaling rates only matings of two carriers are at risk. An additional explanation about the inheritance was added in the introduction (ll. 65-67).

Did the authors take pedigree data into account and test for foaling rates of carrierxcarrier carrierxwild type? This is an important aspect and need clarification.

In ll. 170-171 is now added that we mostly referred to the WFFS status of the sire. An explanation why we focused on the analyses concerning the sires is written in the discussion ll. 372-376. An additional sentence after these lines about the interesting point for further analyses to this topic if the WFFS data would be more complete for the mares was appended (ll. 376-378). 

Line 60-61: Citation missing. It there any evidence confirming the increase of genetic testing?

Citation was added.

Line 365-368: Citation incorrect? In this patent description comments about the earlier ancestor are missing!? Please give details where to find it or correct citation.

Citation was changed.

Reviewer 2:

For their analysis, the authors treat the German Riding horse as a single population. Does this definitely apply for the dataset used? As pointed out in the methods, data from 10 different registries are used. Number of data entries differed significantly between registries (Figure 1). For my knowledge breeding goals, and breeding stock (although

studbook is more or less open) differ between German Warmblood breeds. My question is: Is there population substructure in the dataset? And to what extent does the population substructure/breed registry affect the WFFS allele frequency and hence the outcome of the analysis?

As previous studies have shown there are many crosslinks between the German Warmblood breeds, so that the genetic distance between the single studbooks should not be too far. Especially in modern sport horse breeding there is great exchange between the different breeding associations, not only in Germany. To get an overview for the whole German Warmblood population, which is represented very well in our dataset, we treated the sample of covering data from ten different studbooks as one population. Citation and explanation about this were added in ll. 313-315. The distribution of studbooks within the datasets for analyzing the covering data was similar to the distribution of the whole covering dataset, reflecting the different sizes and breeding numbers of the single studbooks (ll. 232-233, S1 Table). The carrier frequencies per studbook found in our data are now mentioned in ll. 279-281. They were similar to reported frequencies, and highest for Hannover (approx. 13,5%) and Oldenburg (approx. 12%) and lowest for Trakehner (approx. 2,5%). Carrier frequencies for the other studbooks were around 8%. Possible breed differences are considered in the model via the fixed effect of the data provider (every provider of covering data has its own unique database number and if such data provider is working across several studbooks, he has a unique number for each single studbook). This was added in the discussion in ll. 318-320.

Also, for the estimates of association of the WFFS allele and breeding value: the sampling could affect the outcome and should be considered. Association with gait EBV is just detectable in the ‘all’ dataset (Table 2) and it is not in the ‘pub’. What is the difference between ‘all’ and ‘pub’ – with regard to breeds, breeding goals/populations?

The studbook distribution within the all and the pub datasets are similar concerning the ranking of the different studbook proportions. The distribution within these datasets refers mostly to the number of horses within the breeding associations with largest proportions from Hannover and Oldenburg, followed by Trakehner, Oldenburg International and Westfalen. Within the pub datasets for show-jumping there is a slightly higher proportion from Oldenburg International, which could be explained by their breeding focus on show-jumping horses. We suggest that the loss of statistical power from all to pub dataset is mainly caused by the reduced number of horses (due to the publication criteria for national EBVs of stallions). In the pub dataset patterns of results were consistent.

A sentence to the publication criteria was added in ll. 189-190.

An explanation to the studbook distribution within the EBV datasets were added in ll. 298-204.

In the discussion the part to our suggestion that the reduced number of horses caused the loss of statistical power was refined (ll. 426-431).

Hardy Weinberg estimates

(L33; L263). Under the assumed carrier frequency of 9-15%, assumed by the authors, the allele frequency of the minor allele should be 0.045-0.0725. Under Hardy Weinberg equilibrium 0.002-0.005 (0.2-0.5%) homozygous offspring are expected.

Based on an assumed carrier frequency of approx. 9-15% we determined the minor allele frequency. With these minor allele frequencies, we calculated the proportions for AA, Aa and aa with HWE. With the proportions for AA and Aa (which we can see in the population) we made calculations for the mating of a carrier sire with the average mare population:

With minor allele frequencies of 0.05-0.08 the carrier frequencies would be 0.095-0.1472:

Assumption: WFFS carrier frequency of approx. 9% � ~ p(a) = 0.05

~ p(a) = 0.05 � p(A) = 0.95

Hardy-Weinberg

p(AA) = (0.95)² = 0.9025

p(Aa) = 2*(0.95*0.05) = 0.095 (carrier)

p(aa) = (0.05)² = 0.0025

What we can see in the population: 0.9025 + 0.095 = 0.9975 � 1.00

p(AA) = 0.9025/0.9975 = 0.9048

p(Aa) = 0.095/0.9975 = 0.0952

With these values we calculated the scenario for a carrier sire which is mated with the average mare population:

sire(Aa) x dam(AA) = 0.4524 AA

 = 0.4524 Aa

sire(Aa) x dam(Aa) = 0.0238 AA

 = 0.0474 Aa

 = 0.0238 aa

sum = 1.0000

Same calculation was done for the assumed carrier frequency of approx. 15%:

Assumption: WFFS carrier frequency of approx. 15% � ~ p(a) = 0.08

~ p(a) = 0.08 � p(A) = 0.92

Hardy-Weinberg

p(AA) = (0.92)² = 0.8464

p(Aa) = 2*(0.92*0.08) = 0.1472 (carrier)

p(aa) = (0.08)² = 0.0064

What we can see in the population: 0.8464 + 0.1472 = 0.9936 � 1.00

p(AA) = 0.8464/0.9936 = 0.8519

p(Aa) = 0.1472/0.9936 = 0.1481

With these values we calculated the scenario for a carrier sire which is mated to the average mare population:

sire(Aa) x dam(AA) = 0.42595 AA

 = 0.42595 Aa

sire(Aa) x dam(Aa) = 0.03703 AA

 = 0.07405 Aa

 = 0.03703 aa

sum = 1.00000

So, we expected between 2.4-3.7% more foal losses from a mating carrier sire to the average mare population.

The sentence about our calculation was specified in l. 242 and 283 and also in the discussion (ll. 370-371): We calculated the expected proportion of homozygous offspring from a carrier sire using HWE and the assumed carrier frequency of 9-15% within the riding horse population.

Furthermore, it has recently been shown that WFFS allele frequency differs between Warmblood breeds/registries (doi: 10.3390/genes11121518), and seems to be highest in Hanoverians (doi: 10.1111/evj.13271). See above comment.

It is surely true that difference of WFFS allele frequencies exist between breeds (or: studbooks). And our findings, now included in the results section, are consistent with figures from the literature (ll. 279-281).

Scenarios underlying the significant impact of WFFS allele on foaling rate

The effect of the WFFS allele on foaling rates is rather minor, although significant. I would appreciate a comprehensive discussion on the scenarios that could lead to this observation.

The determined effect of the WFFS mutation is small but with 2.7% its order fits very well to the expected proportion of homozygous offspring from a carrier sire mated to the average mare population with a carrier frequency of 9-15%. Discussion of the effect size has been amended to clarify the reason for the relatively small effect (ll. 369-371).

The significant impact of WFFS allele on dressage performance

Also for this fact, a broader discussion on the possible causes (WFFS causative, hitchhiking with a linked locus or even just drift?) that could lead to the association of the WFFS allele with dressage performance would be beneficial.

The authors assume that there may be favorable aspects linked with the mutation (ll. 406-409) and a kind of involuntary co-selection for WFFS (hitchhiking). Currently the data for distinguishing between the possible reasons for the impact of WFFS on dressage performance is missing, so that studies to further investigate this topic are needed (ll. 439-442).

The title does not fully reflect the content of the article

The idea behind the title was to reflect the main topic of our study. To investigate possible reasons for the long survival of this genetic defect, we added the analyses of variance with the breeding values to clarify if there is an association with performance traits. The title was not adjusted.

L27 ..also in respect ‘to other traits’. Not precise.

Changed to ‘also in respect to performance traits’ (line 27).

L33. … Hardy Weinberg principle implied an expected difference of 2.4-3.7% in the foaling rates of carrier and noncarrier stallions … This calculation is for me not clear (see major comments)

No changes were made because checking of our calculations did not indicate errors. See reply to the major comment concerning the HWE calculations.

The WFFS mutation which is central in this article should be described in more detail (Genome coordinates, OMIA link) Further the mode of inheritance, which is autosomal recessive - should be pointed out in the introduction.

The mode of inheritance has been further specified as autosomal recessive (l. 63), followed by detailed explanation what this means. Genome coordinates and OMIA link have been added (ll. 65-67).

L161-163 fits better to section covering data.

These lines were meant as legend to figure 2. The new line behind the figure title was removed, that the position of the legend now corresponds correctly to the guidelines of the journal.

L229 allele frequency was also estimated in this work – and it was lower, why not refer to this as well?

The frequency found in our data (mentioned in l. 284) is similar to the frequencies found in previous studies (at the lower end). We preferred to orientate our analyses to all available facts to the WFFS carrier frequencies from previous studies. 

L243 more explanation is necessary for the describing all/pub data (maybe supplementary Table).

Explanation on the publication criteria (ll. 189-190) and information on the distribution of studbooks within these datasets (ll. 201-204) was added directly in the text.

Table1: A supplementary table with detailed information on registries/breeds enclosed in the respective restriction sets would be helpful.

The distribution of studbooks within the datasets described in table 1 are very similar to the distribution in the whole covering data set, shown in figure 1. A sentence about this was added just before table 1 (ll. 232-233). A supplementary table (S1 Table) was added.

Table2: More detailed description on breeding goals (discipline) and registries for the horses would be helpful. Were all EBVs determined in the same individuals? How can one explain loss of significance in Dressage values when only pub data are analysed?

Probably table 3 was meant. More information about the breeding goals of the studbooks and their distribution within the EBV datasets was added in ll. 198-204. 

Yes, all EBVs were determined for all horses within the genetic evaluation. 

The loss of significance using the pub data is discussed in lines 426-431 where the authors assume that the major impact on the statistical power is due to the reduced number of horses within the pub datasets. 

L334 This conclusion is not clear to me.

Probably originally l. 434 within the conclusion part was meant. It is challenging to determine a cause for heaving “no foal”, so that the authors think that 2.7% lower foaling rates in carrier sires compared to WFFS free ones are definitely a relevant finding of this study.

---

## [Decision Letter · Decision Letter 1]

20 Apr 2022

Quantifying the effect of Warmblood Fragile Foal Syndrome on foaling rates in the German riding horse population

PONE-D-21-32388R1

Dear Dr. Wobbe,

We’re pleased to inform you that your manuscript has been judged scientifically suitable for publication and will be formally accepted for publication once it meets all outstanding technical requirements.

Kind regards,

Sylvie Mazoyer, Ph.D.

Academic Editor

PLOS ONE

Additional Editor Comments (optional):

Reviewers' comments:

Reviewer's Responses to Questions

**Comments to the Author**

1. If the authors have adequately addressed your comments raised in a previous round of review and you feel that this manuscript is now acceptable for publication, you may indicate that here to bypass the “Comments to the Author” section, enter your conflict of interest statement in the “Confidential to Editor” section, and submit your "Accept" recommendation.

Reviewer #1: All comments have been addressed

Reviewer #2: All comments have been addressed

2. Is the manuscript technically sound, and do the data support the conclusions?

Reviewer #1: Yes

Reviewer #2: Yes

3. Has the statistical analysis been performed appropriately and rigorously? 

Reviewer #1: Yes

Reviewer #2: N/A

4. Have the authors made all data underlying the findings in their manuscript fully available?

Reviewer #1: Yes

Reviewer #2: Yes

5. Is the manuscript presented in an intelligible fashion and written in standard English?

Reviewer #1: Yes

Reviewer #2: Yes

6. Review Comments to the Author

Reviewer #1: The manuscript by Wobbe et al. has been significantly improved by the revisions. I accept the manuscript in its current stage.

Reviewer #2: I explicitly thank the authors for the clarification with the problems I had interpreting their expected WFFS genotype frequencies. It is now unambiguously explained in the article that their calculations are based on the condition of mating with a carrier stallion, which I didn't get in the previous version. I fully agree with the authors' calculations and their interpretation and appreciate the added explanatory words in the text.

I found a few minor spelling errors, and would recommend a language check before publication.

7. PLOS authors have the option to publish the peer review history of their article (what does this mean?). If published, this will include your full peer review and any attached files.

Reviewer #1: No

Reviewer #2: No

---

## [Editor Report · Acceptance letter]

25 Apr 2022

PONE-D-21-32388R1 

Quantifying the effect of Warmblood Fragile Foal Syndrome on foaling rates in the German riding horse population 

Dear Dr. Wobbe:

I'm pleased to inform you that your manuscript has been deemed suitable for publication in PLOS ONE. Congratulations! Your manuscript is now with our production department. 

Kind regards, 

on behalf of

Dr Sylvie Mazoyer 

Academic Editor

PLOS ONE